# Analysis of Risk Factors for Phonation Disorders after Thyroid Surgery

**DOI:** 10.3390/biomedicines10092280

**Published:** 2022-09-14

**Authors:** Mateusz Głód, Dominik Marciniak, Krzysztof Kaliszewski, Krzysztof Sutkowski, Jerzy Rudnicki, Marek Bolanowski, Beata Wojtczak

**Affiliations:** 1Department of General, Minimally Invasive and Endocrine Surgery, Wroclaw Medical University, Borowska Street 213, 50-556 Wroclaw, Poland; 2Department of Dosage Form Technology, Wroclaw Medical University, Borowska Street 211 A, 50-556 Wroclaw, Poland; 3Department of Endocrinology, Diabetes and Isotope Therapy, Wroclaw Medical University, Pasteura Street 4, 50-367 Wroclaw, Poland

**Keywords:** thyroid surgery, vocal cord paralysis, complications, risk factors

## Abstract

Phonation disorders after thyroidectomy are among the most common complications and occur as a result of recurrent laryngeal nerve (RLN) injury. The multivariate analysis of risk factors for phonation disorders after thyroidectomy was assessed. A group of 830 patients with 1500 RLNs at risk of injury during thyroidectomy were analyzed retrospectively. The impact of the method of RLN identification, age, sex, BMI, kind of thyroid surgery, pathology, surgeon’s experience and thyroid volume on vocal cord paralysis was analyzed. We found that the retrosternal goiter and the volume above 100 mL were the most important risk factors for both transient and permanent paralysis. Thyroid cancer had a statistically significant impact on the increase in permanent paralysis, while this indication had practically no impact on transient paralysis. Among patients over 65 years with obesity, the probability of transient complications approximately doubled, with no effect on the permanent paralysis. Men were approximately 1.7 times more likely to develop any type of phonation disorder. Secondary operations more than doubled the risk of transient and permanent vocal cord paralysis. Thyroidectomy with only visual RLN identification was associated with a risk of both transient and permanent vocal cord paralysis almost two times higher, compared to neuromonitoring.

## 1. Introduction

The human voice is a basic instrument for functioning in society. It facilitates unhindered communication, expression of thoughts and emotions and the formation of interpersonal relationships. It is the basic working tool for people who use their voices professionally: actors, teachers and singers.

In the last three decades, the number of thyroid surgeries has tripled; currently, it is one of the most frequently performed procedures in general surgery. These operations, like any other surgical procedure, are associated with a risk of complications, including phonation disorders. About 1 in 10 patients experience transient voice disorders after thyroid surgery, and 1 in 25 experience permanent voice disorders [1]. The main cause of phonation disorders after thyroid surgery is injury of the recurrent laryngeal nerve (RLN), due to its proximity to the thyroid gland. This nerve is responsible for the innervation of all the muscles of the larynx except the cricothyroid muscle [2,3,4,5,6]. The range of clinical symptoms in the case of RLN damage is very wide, from slight changes in the timbre of the voice and hoarseness in the case of unilateral damage to the RLN, to complete silence, dyspnea and stridor in cases of bilateral RLN injury, which can be a life-threatening condition requiring a tracheostomy. Jeannon et al., based on a meta-analysis of 27 articles involving a total of 25,000 patients, showed the percentage of vocal cord paralysis as 9.8%, of which 2.3% was permanent [1].

On the other hand, phonation disorders in the course of injury to the external branch of the superior laryngeal nerve (EBSLN) are the most underestimated complication following thyroid surgery [4,7,8,9]. This nerve, a branch of the vagus nerve, contains motor fibers that innervate the cricothyroid muscle and is responsible for regulating the tension of the vocal folds. As a result of EBSLN damage, the vocal fold becomes flaccid, reducing the volume of the voice and manifested by an inability to produce high tones, which is especially important for people who work with their voices. In addition, the timbre of the voice may change. Bilateral EBSLN damage results in a monotonous, hoarse voice that diminishes during vocalization. A consequence of bilateral EBSLN damage is disturbance of sensation in the upper half of the larynx, which leads to swallowing disorders and choking when swallowing [3,7,10]. Voice quality studies after thyroid surgery indicate that this nerve may be damaged in up to 60% of patients [1,7,11].

The aim of this study was to assess the risk of vocal fold paralysis following thyroid surgery, and to identify risk factors that have a statistically significant impact on increases in the probability of their occurrence. Risk factors include those caused by pathologies of the thyroid gland itself, as well as those resulting from the operating procedure used, the surgeon’s lack of experience or the availability of neuromonitoring during thyroid surgery. Demographic factors as risk factors were also analyzed.

## 2. Materials and Methods

### 2.1. Study Population

On 13 May 2020, the consent of the Bioethics Committee of the Medical University of Wrocław (Poland) to conduct the study was obtained (KB-280/2020). The medical records of patients treated surgically for various diseases of the thyroid gland at the Department of General, Gastroenterological and Endocrine Surgery at the Medical University of Wrocław from 2011 to 2015 were analyzed retrospectively. The study included 830 patients (691 women and 139 men) in whom 1500 recurrent laryngeal nerves were at risk of damage (RLN at risk) during thyroid surgery. The patients’ demographics and clinical characteristics are presented in Table 1.

### 2.2. Data Collection

In order to take into account the complex specificity of postoperative complications in the planned statistical analyses, three separate endpoints (Vz1–Vz3) were defined, which served as dependent variables in the computational methods employed. The total number of vocal cord paralysis (Vz1) cases after thyroid surgery was 57 (3.8%), calculated by the number of RLNs at risk of injury (1500 RLNs at risk): there were 37 transient paralysis (TP/Vz2) cases (4.5%) and 20 permanent paralysis (PP/Vz2) cases (1.3%). Unilateral paresis occurred in 49 (5.9%) patients and bilateral paresis in 4 (0.5%) patients.

Before thyroid operation all the patients underwent ENT examination of their vocal cords (indirect examination or videolaryngoscopy). The same examination was also performed on the first or second postoperative day. In cases of abnormal vocal cord mobility, videostroboscopy was performed to confirm paresis or paralysis of the vocal cords and to assess the glottis width. The examination of vocal cords was repeated after 1, 3 and 6 months in cases of proven paralysis. Transient paralysis was defined as paralysis that resolved within 6 months after surgery; if the paralysis persisted for 6 months it was classified as permanent.

The influence of 17 different factors (Vzn1–17) with potential impact on the risk of complications is analyzed in Table 2.

### 2.3. Statistical Analysis

Univariate, multivariate and multidimensional statistical procedures (available in the statistical software (Tibco Software Inc., Palo Alto, CA, USA) Statistica^®^ Pl version 13.3 by StatSoft) were used in the study, enabling us to analyze dichotomous dependent variables. These methods use three completely different computational techniques. The chi2 test and meta-analysis are based on the assessment of statistical significance of the probability that the analyzed fraction of the dichotomous variable may come from a population with a normal distribution of the tested attribute. Logistic regression using univariate and multivariate linear regression techniques is based on the general linear model (GLM) of Gauss and Markov. Principal component analysis (PCA) and correspondence analysis—the most computationally sophisticated techniques used in the study—use matrix decomposition according to singular values, allowing us to determine eigenvectors and eigenvalues, replacing the classic method of determining the correlation matrix in a “peer-to-peer” comparison scheme.

For each of the variables analyzed, the developed meta-analysis model determined the odds ratio (OR), its statistical significance (*p*), 95% confidence interval and the percentage each variable contributed to the explanation of the alterations of variance of the model. Using the assumptions of the meta-analysis model of fixed effects, the value and statistical significance of the total OR resulting from an interaction between the analyzed variables were calculated. In all of the statistical analyses, the significance level was α = 0.05.

## 3. Results

The size of the sample subjected to statistical analysis was *n* = 1500, in which various types of complications affected 3.8% (*n* = 57). An analysis of test power for this sample size showed that given the assumption that postoperative complications in the entire population of all procedures of this type average from 2.0% to 5.0%, the minimum sample size ensuring that the expected power will not be lower than 0.9 (β ≥ 0.9) is Nmin = 414. As the sample in our study is approximately 3.6 times larger than this Nmin, we can assume that the conclusions of our statistical analyses should be highly convincing.

In the first stage of the statistical analyses, the general relationships between all the variables analyzed were assessed using multidimensional data mining techniques, based on reducing the number of dimensions in the principal components analysis (PCA) and correspondence analysis to two (PC_1 and PC_2). Analyzing the PC_1 and PC_2 eigenvector charge scattering plots initially allowed us to correlate the occurrence of postoperative complications with the following factors: the presence of a retrosternal goiter, visual RLN identification, total volume above 100 mL, secondary surgery, indications of Graves’ disease (GD) and thyroid cancer, a longer duration of surgery, total thyroidectomy, BMI values for obesity, male gender and older age. The multidimensional relationships among risk factors for complications and the total number of cases of vocal cord paralysis based on the data mining technique are presented in Figure 1.

### 3.1. Paralysis of the Vocal Cords Immediately after Thyroid Surgery

Our analysis with Pearson’s non-parametric chi2 test showed statistically significant correlations between the occurrence of paralysis of the vocal folds in the immediate postoperative period and the following:The presence of a retrosternal goiter (chi2 = 15.1; *p* = 0.0001, OR 2.8, 95%CI 1.63–4.82). In the group of patients with early vocal cord paralysis, the percentage of patients with a retrosternal goiter was 42.1%; in the group of patients without complications, this percentage constituted only 20.6%. An odds ratio (OR) value of 2.8 indicates that the presence of a retrosternal goiter increases the probability of early vocal cord paralysis almost three-fold.Whether the operation was primary or secondary (chi2 = 4.9; *p* = 0.027, OR 2.25, 95%Cl: 1.34–3.77). In the group of patients with early vocal cord paralysis, the percentage of patients for whom it was secondary surgery was 15.8%, and in the group of patients without paralysis it was only 7.7%. In the group of patients for whom it was secondary surgery, the probability of complications was about 2.3 times higher than in the group of patients for whom it was primary surgery (OR = 2.25).The indication for surgery (chi2 = 14.5; *p* = 0.0023). In the group of patients with early vocal cord paralysis, the percentage of patients for whom the indication for surgery was thyroid cancer was 23.0%. The probability of vocal cord paralysis in the group of people with this indication was about 3.4 times higher than in the group of patients with other indications (OR = 3.38, 95% Cl: 1.721–6.651) according to the logistic regression model.

The following variables had no statistical significance: neuromonitoring not used during the surgery, age over 65 years vs. 65 and under, female vs. male.

The conclusions resulting from the analyses based on Pearson’s chi2 statistics are consistent with the results of modeling carried out using the logistic function. The one-way logistic regression models we constructed confirmed the statistical significance of correlations between early vocal cord paralysis (Vz1) and the following variables: Vnz4—primary operation vs. secondary operation, Vnz9—the indication for surgical treatment (clinical diagnosis), Vnz6—retrosternal goiter.

Additionally, the one-way statistical analysis of logistic regression showed that the variable Vnz13—total goiter volume above 100 mL—had statistically significant influence on increasing the probability of early vocal cord paralysis. In the group of patients with the highest goiter volume (V > 100), early vocal cord paralysis was 3.3 times higher than in the other groups (OR = 3.34, 95%Cl: 1.483–7.543).

One-way logistic regression analyses performed for the variables on the quotient scales showed they had a small but statistically significant impact on the probability of early vocal cord paralysis: age (years) (OR = 1.06, *p* < 0.05, 95%Cl: 1.057–1.068); operation time (min) (OR = 1.06, *p* <0.05, 95%Cl: 1.054–1.064); BMI (raw score) (OR = 1.13, *p* <0.05, 95%Cl: 1.12–1.14); total volume (mL) (OR = 1.08, *p* < 0.05, 95%Cl: 1.074–1.089).

Using the available stepwise regression algorithms, the essence of a statistically multivariate, predictive logistic regression model was also developed, taking four variables into account simultaneously: Vnz1 (neuromonitoring = 1/visualization = 0), Vnz13 (total goiter volume >100 mL), Vnz6 (retrosternal goiter), Vnz9 (indication for surgical treatment).

The results of the meta-analysis carried out according to the variable effects model confirmed the conclusions of all the statistical analyses performed so far; the results are summarized in Figure 2 and Table 3

The meta-analysis showed the statistically significant impact of five variables on increases in the probability of the total number of vocal cord paralysis occurrences. These are: secondary surgery (secondary surgery = 1/primary surgery = 0; (OR = 2.25; *p* = 0.0312, 95%Cl: 1.08–4.71%)), total volume (V > 100 = 1/V ≤100 = 0; (OR = 3.11; *p* = 0.0046, 95%Cl: 1.42–6.81)), retrosternal goiter (retrosternal goiter present = 1/retrosternal goiter absent = 0; (OR = 2.81; *p* = 0.0002, 95%Cl: 1.63–4.82%)), duration of surgery (t > median (= 55.0) = 1/t ≤ median (= 55.0) = 0; (OR = 2.00; *p* = 0.0132, 95%Cl: 1.16–3.46)), diagnosis of carcinoma (yes = 1); (OR = 2.98, *p* = 0.0009, 95%Cl: 1.57–5.68)).

An interaction model assuming the effects of the variables showed that the odds ratio resulting from the simultaneous impact of all 11 variables included in the meta-analysis on increases in the probability of the total number of cases of vocal cord paralysis is OR = 1.83 (95%Cl 1.44–2.33), and is highly significant statistically, with *p* = <0.0001.

### 3.2. Transient Paralysis of the Vocal Cords

Our analysis with Pearson’s non-parametric chi2 test showed a statistically significant correlation between the occurrence of transient paralysis of the vocal cords and the following:The presence of a retrosternal goiter (chi2 = 8.3; *p* = 0.0041, OR 2.57, Cl: 1.32–5.03). In the group of patients with transient paralysis of the vocal cords, the percentage of patients with retrosternal goiters was 40.5%; in the group of patients without paralysis, this percentage was only 20%. The OR value of 2.57 indicates that the presence of retrosternal goiter increases the probability of transient paralysis of the vocal cords 2.5-fold.The age of the patient (age ≤65 years/age >65; chi2 = 5.2; *p* = 0.02, OR 2.2, 95%Cl: 1.1–4.34). In the group of patients with transient vocal cord paralysis, the percentage of patients over 65 years in age was 35%, and in the group of patients without complications, this percentage was only 20%. The OR value of 2.2 indicates that age above 65 years more than doubles the probability of transient paralysis of the vocal cords.

The conclusions resulting from analyses based on Pearson’s chi2 statistics are consistent with the results of modeling carried out with the use of the logistic function. The one-way logistic regression models we constructed confirmed the statistical significance of the correlations between the occurrence of transient paralysis of the vocal cords (Vz2) and the following variables: Vnz2 (age ≤65/>65) and Vnz6 (retrosternal goiter).

A one-way logistic regression analysis performed for variables on quotient scales according to the model with the intercept β0 = 0 showed no statistically significant influence of age, operation time, BMI or total volume on the probability of transient paralysis of the vocal cords.

Using the available stepwise regression algorithms, a statistically significant, multivariate, predictive logistic regression model was developed, taking into account two variables simultaneously: Vnz2 (age ≤65/>65) and Vnz6 (retrosternal goiter).

The results of the meta-analysis carried out according to the variable effects model confirmed the conclusions of all the statistical analyses performed so far. The results are summarized in Figure 2 and Table 3

The meta-analysis showed the statistically significant influence of two variables on increases in the probability of transient paralysis of the vocal cords. These are: age (>65 = 1/ ≤65 = 0; OR = 2.18; *p* = 0.026 95%Cl: 1.10–4.34) and retrosternal goiter (present = 1/absent = 0; OR = 2.58; *p* = 0.0055 95%Cl:1.32–5.03).

The interaction model assuming the effects of the variables showed that the odds ratio resulting from the simultaneous impact of all 11 variables included in the meta-analysis on increases in the probability of early transient paralysis of the vocal cords is OR = 1.49 (95%Cl: 1.16–1.93), and is highly significant statistically at *p* = 0.0019.

### 3.3. Permanent Paralysis of the Vocal Cords

Our analysis with Pearson’s non-parametric chi2 test showed a statistically significant correlation between the occurrence of permanent vocal cord paralysis and the following:The presence of a retrosternal goiter (chi2 = 6.77; *p* = 0.009, OR 3.06, 95%Cl: 1.26–7.46). In the group of patients with permanent paralysis of the vocal cords, the percentage of patients with retrosternal goiter was 45%, and in the group of patients without paralysis, this percentage amounted to only 21%. The OR value of 3.06 indicates that the presence of a retrosternal goiter increases the probability of permanent vocal cord paralysis almost three-fold.Whether the operation under consideration was primary or secondary (chi2 = 3.97; *p* = 0.046, OR 2.94, 95%Cl: 0.97–8.94). In the group of patients with permanent vocal cord paralysis, the percentage of patients for whom it was secondary surgery was 20%; in the group of patients without complications, it was only 7.8%. In the group of patients for whom the surgery was secondary, the probability of permanent paralysis was approximately three times higher than in the group of patients for whom it was primary surgery (OR = 2.94).The indication for surgery (chi2 = 31.5; *p* = 0.00000). In the group of patients with permanent vocal cord paralysis, the percentage of patients for whom the indication for surgery was thyroid cancer was 35.0%. The probability of paralysis in the group of people with this indication was about 10 times higher than in the group of patients with other indications (OR = 10.71, 95%Cl: 3.351–34.202) according to the logistic regression model).The presence of a single or multiple focal lesions in the thyroid vs. parenchymal goiter (chi2 = 28.25; *p* < 0.00001). In the group of patients with permanent vocal cord paralysis, the percentage of patients with a parenchymal goiter was 20.0%. The probability of complications in the group of patients with this type of lesion was about four times higher than in the group of patients with single or multiple tumors (OR = 3.77, 95%Cl: 1.055–13.507) according to the logistic regression model.Goiter volume (chi2 = 8.82; *p* = 0.012). In the group of patients with permanent cord paralysis, the percentage of patients with large goiters (>100 mL) was 20.0%. The probability of complications in the group of people with this size goiter was about five times higher than in the group of patients with goiters up to 100 mL (OR = 5.02, 95%Cl: 1.538–16.387) according to the logistic regression model.

The conclusions resulting from our analyses based on Pearson’s chi2 statistics are consistent with the results of modeling carried out using the logistic function. The one-way logistic regression models we constructed confirmed the statistical significance of the correlations between permanent vocal cord paralysis (Vz3) and the following variables: Vnz9 (the indication for surgery), Vnz6 (retrosternal goiter), Vnz10 (a single tumor = 0/bilateral nodules = 1/parenchymal goiter = 2), Vnz13 (the total volume of goiter).

Our one-way statistical analysis of logistic regression showed that the influence of the variable Vnz4 (whether the operation was primary or secondary) was on the threshold of statistical significance.

The one-way logistic regression analysis performed for variables on quotient scales, carried out according to the model with the intercept β0 = 0, showed that the total volume of the goiter (OR = 1.014; *p* = 0.0038, 95%Cl: 1.004–1.023) had a statistically significant effect on the probability of permanent vocal cord paralysis. In the case of permanent vocal cord paralysis, it was not possible to develop a statistically significant multifactor predictive logistic regression model using the available stepwise regression algorithms. On the other hand, the multivariate logistic regression analysis carried out for variables on quotient scales carried out according to the model with the intercept β0 = 0 showed that the following variables had statistically significant effects on the probability of permanent vocal cord paralysis: BMI (OR = 0.87; *p* < 0.001, 95%Cl: 0.823–0.926), the age of the patient (OR = 0.972; *p* = 0.044, 95%Cl: 0.945–0.999), total goiter volume (mL) (OR = 1.014; *p* = 0.004, 95%Cl: 1.005–1.024). The results of the meta-analysis according to the variable effects model are presented in Figure 2 and Table 3.

### 3.4. Meta-Analysis

The meta-analysis showed the statistically significant influence of five variables on increases in the probability of permanent vocal cord paralysis:Secondary surgery (secondary surgery = 1/primary surgery = 0; (OR = 2.94; 95%Cl: 0.97–8.94, *p* = 0.05): the result is on the verge of statistical significance);Total volume (mL) (V > 100 = 1/V ≤ 100 = 0; (OR = 4.62; 95%Cl: 1.51–14.15, *p* = 0.0074))Retrosternal goiter (present = 1; (OR = 3.06; 95%Cl: 1.26–7.46, *p* = 0.0132));Duration of operation (t > median (55 min) = 1/t ≤ median = 0; (OR = 1.79; 95%Cl: 1.03–7.03, *p* = 0.009));Diagnosis of carcinoma (yes = 1; OR = 5.32, 95%CL: 2.09–13.56, *p* = 0.0005)).

The interaction model assuming the effects of variables showed that the odds ratio resulting from the simultaneous impact of all 11 variables included in the meta-analysis on increases in the probability of permanent vocal cord paralysis is OR = 2.0, 95%Cl: 1.32–3.04, and is highly significant statistically (*p* = 0.0011).

In order to summarize the results obtained and to clearly visualize the conclusions of the conducted research, a meta-analysis of variable effects was carried out according to a scheme that takes into account all the possible interactions among the subgroups identified. This approach allowed us to assess the differences in the degrees of correlation resulting from a given endpoint. The results of this analysis are shown in Figure 2 and Table 3.

## 4. Discussion

Phonation disorders, which are most often the result of damage to the recurrent laryngeal nerve or the external branch of the superior laryngeal nerve during thyroid surgery, significantly reduce the quality of life of patients and may cause permanent disability. Hence, the assessment of phonation disorders after thyroid surgery along with an attempt to identify factors that may significantly affect this complication is an important subject of research.

In our cohort, 2.5% were cases of temporary vocal cord paralysis and 1.3% were instances of permanent paralysis. These results are within the range of vocal cord paralysis following thyroid surgery described in the literature, where the frequency of transient paralysis is estimated from 1.4% to 38.4% (average 9.8%), and permanent paralysis from 0 to 18.6% (average 2.3%). These results were based on a review of 27 articles, including 25,000 patients operated on for various thyroid diseases [1]. The percentage of vocal cord paralysis in our research group is similar to the results published by Thomusch et al. [12], where transient paralysis accounted for 2.1%, and permanent paralysis 1.1%, and the results of Rosato et al. [13], where the percentage of temporary paralysis was 3.4%, and permanent paralysis was 1.4% in a group of almost 15,000 patients. In 2019, Hung-Chun Chen et al. presented a similar percentage of vocal cord paralysis, amounting to 2.1% of 2815 patients who underwent thyroidectomy [14]. However, the reported results of vocal cord paralysis should be approached quite cautiously. As shown by a Scandinavian study, the frequency of vocal cord paralysis was 4.3%; at the same time, the number of cases in individual centers was closely related to the percentage of laryngological examinations performed after thyroid surgery. In centers where laryngological examinations were routinely performed after thyroid surgery, the frequency of paralysis was twice as high as when examinations were performed only in patients with audible voice disorders [11]. In order to fully assess the effects of thyroid surgery on vocal cord mobility, laryngological examinations both before and after surgery should be the standard of care [15]. The literature indicates that the percentage of laryngological tests before thyroid gland surgery is only 6.1–54% [1,16], and in many centers the number is completely unknown. Moreover, considering the frequency of vocal cord paralysis after thyroid surgery without taking into account whether they were primary or secondary operations, or related to a group of benign diseases or thyroid carcinomas, is very generalized. Furthermore, the introduction of recurrent laryngeal nerve monitoring in thyroid surgery and the standardization of recurrent laryngeal nerve neuromonitoring techniques in 2011 [4], as well as the external branch of the superior laryngeal nerve in 2013 [7], significantly influenced the results of phonation disorders following thyroid surgery. Hence, specifying the risk factors for phonation disorders after thyroid surgery seems to be particularly important in practical terms for both the surgeon and the patient.

The statistical analyses we performed showed the particularly significant influence of two factors on increases in the probability of vocal cord paralysis: the presence of a retrosternal goiter and thyroid gland volume exceeding 100 mL. The probability of transient paralysis of the vocal cords in patients with retrosternal goiters is 2.6 times higher, and the probability of permanent paralysis is 3.1 times higher than in the group without this feature. Similarly, a patient with a thyroid volume exceeding 100 mL is 2.2 times more likely to suffer from transient paralysis and as much as 4.6 times more likely to suffer from permanent paralysis. The odds ratios calculated for these two variables, taking into account the interactions between our three different endpoints, are OR = 2.8 for the variable “retrosternal goiter” and OR = 3.1 for the variable “volume above 100 mL”. Both of these values are highly significant statistically. Similar results were obtained by Vetshev et al., who, on the basis of 1272 thyroid surgeries, considered the size of the thyroid and the presence of a retrosternal goiter to be among the most important risk factors for transient and permanent paralysis of the vocal cords [17]. How strong a risk factor a retrosternal goiter is for paralysis of the vocal cords was demonstrated by a multicenter study by Testini et al., during which nearly 15,000 patients with retrosternal goiters underwent clinical observation. Permanent and transient paralysis of the vocal cords was significantly more frequent among patients operated on for retrosternal goiters (both cervical and thoracic) than among patients operated on for goiters in a typical location on the neck (*p* < 0.001); the highest percentage of complications was in the group of patients operated on through the thoracic approach [18]. Testini et al. also compared thyroid operations performed only through the cervical approach, and in this group of patients the percentage of paralysis of the vocal cords in cases of retrosternal goiters was significantly higher than in the group of patients with goiters in a typical localization on the neck (*p* < 0.001) [18]. The authors of most publications on retrosternal goiters agree that the risk of complications in this group is always greater than in cases of cervical goiters [19,20,21]. Interestingly, in their study Wei Li et al. specified risk factors influencing increases in the percentage of paralysis of the vocal cords in cases of retrosternal goiters; these were operations due to recurrent goiters and the location of the thyroid tissue near the manubrium of the sternum. In this group of patients, the percentage of permanent complications was as high as 14% [22]. Goiter surgery is a challenge even for very experienced surgeons, and it always requires appropriate preparation of the patient and the awareness that about 2% of these procedures will require opening the chest [22]. In addition, it seems advisable to refer patients for thyroid surgery at a much earlier stage of the disease, in order to prevent retrosternal localization or enlargement of the goiter.

When the indication for surgery is thyroid cancer, it has a very significant impact on increases in the probability of permanent vocal cord paralysis, which is 5.3 times higher in this group than in the group with other indications. It should be noted here that this indication has practically no effect on the probability of transient paralysis of the vocal cords (OR = 1.0). The significantly worse treatment outcomes among the patients with thyroid cancer in our research group can be explained by the relative inexperience of the clinic’s team, especially in cases of thyroid reoperations due to recurrences of thyroid cancer or radicalization due to thyroid tissue originally left in the postoperative bed—an operation that is technically extremely difficult, requiring the most extensive surgical experience. Such patients are normally referred to higher-level centers. Similarly, in a study by Caulley et al. [23], thyroid cancer turned out to be the most important risk factor for phonation disorders. A study by Heikknen et al. involving 920 operations showed that thyroid cancer increased the risk of complications three-fold, and that among all the factors, it was the most significant [24]. An analysis of risk factors for complications conducted by Gunn et al. in 2020 based on a group of 11,000 patients showed that thyroid cancer was a statistically significant factor in vocal cord paralysis (OR = 2.1; *p* < 0.001) [25].

Age exceeding 65 years and a BMI value indicating obesity are two characteristic and non-obvious risk factors for postoperative paralysis. Both of these factors approximately double the probability of transient paralysis of the vocal cords but have no effect on the frequency of permanent paralysis. The odds ratio calculated for these two factors, taking into account interactions between the dependent variables, is statistically significant, although it is only 1.7. Age over 60 years was identified as a risk factor for vocal cord paralysis in a multifactorial analysis of complications by Hung-Chun Chen et al., who obtained almost identical results as in the present analysis, indicating that the relative risk of vocal cord paralysis in patients over 60 years of age was twice as high as in the group of younger patients [14]. Very similar results concerning the influence of age on complications were presented by Gunn et al., who showed that in patients over 65 years of age the risk of paralysis was 1.6 times higher than in younger patients [25]. Interestingly, the study by Chen et al. proved the relationship between the occurrence of the external branch of the superior laryngeal nerve injury and an increased incidence of diabetes in these patients. Perhaps it is the increase in the prevalence of diabetes and other comorbidities in the elderly, as well as the frequent use of multiple drugs such as antithrombotic or antiplatelet drugs in the group of patients over 65 years of age, which makes their laryngeal nerves especially prone to damage. This could explain our results, which showed that age was significant for the occurrence of transient but not permanent paralysis. Permanent paralysis is most often the result of irreversible mechanical damage during thyroid surgery [24].

The inclusion of body mass index (BMI) and obesity in particular as a risk factor in this dissertation seems to be justified for two reasons. In its 2016 report, the World Health Organization (WHO) noted that the number of obese people has increased approximately three-fold since 1975. In addition, in recent years, more and more patients with high BMIs are undergoing thyroid surgery, which is most likely due to the higher prevalence of thyroid cancer in this group [26,27,28]. The published results for body mass index or obesity as a risk factor for vocal cord paralysis after thyroid surgery are not consistent. In our research group, a BMI in the obese range approximately doubled the number of cases of transient paralysis of the vocal cords. On the other hand, the results of a retrospective study by Gian Luigi Canu et al. did not confirm our observations; they found no statistically significant differences (*p* = 1.0) between the number of paralysis cases (2.8%) in patients with BMIs up to 30 kg/m^2^ and in those with BMIs over 30 kg/m^2^ (2.22%) [28]. A publication by Tresallet et al. based on the observation of 1216 patients treated surgically for thyroid cancer and taking BMI into account showed a higher risk of recurrent laryngeal nerve palsy in obese patients, but the difference was not statistically significant [29].

Gender also has a slight but constant influence (independent of the endpoint adopted) on the probability of vocal cord paralysis. Our analyses show that men are approximately 1.7 times more exposed than women to the risk of any type of vocal cord paralysis. The odds ratio calculated for the gender variable, taking into account interactions between the dependent variables, is OR = 1.7, and is statistically significant. Our clinical, but subjective experience indicates that thyroid surgery performed on men can be technically more difficult than on women. Perhaps this is due to the fact that men usually present for surgery much later than women, which may be associated with overlapping other factors: the presence of retrosternal goiters and larger goiters. In the literature, we find few studies indicating gender as a risk factor for vocal cord paralysis after thyroid surgery. Hermann et al. [30], based on a retrospective analysis of 10 years of personal experience, showed that women were more prone to paresis after thyroid surgery than men (5.6% vs. 2.9%; *p* = 0.001). Similar results were presented by Thomusch et al. [12], who indicated female sex as a significant risk factor for complications, almost doubling the percentage of RLN injuries. Thomusch et al. suggested that the more delicate anatomy of the surrounding tissues in women and the greater consumption of aspirin by women could cause problems with maintaining the proper hemostasis necessary to identify the RLN.

In our study, the values of the odds ratios taking into account interactions among the results obtained for all three analyzed endpoints were statistically significant for visual identification (OR = 1.7; *p* = 0.02) and secondary surgery (OR = 2.25; *p* = 0.0022).

Recurrent goiters in thyroid surgery are a topic that has been reported by many authors [2,24,31]. Due to the altered anatomy of the course of the laryngeal nerve after primary surgery, its identification is extremely difficult in these cases; moreover, this nerve may be incorporated into the postoperative scar and can be difficult to distinguish from other tissues. This group of operations includes both operations performed for the recurrence of a goiter many years after primary (usually subtotal) thyroid surgery, as well as operations for local recurrences of thyroid cancer or originally non-radical surgery. The present study confirms the findings of other studies, that recurrent goiter is a risk factor for phonation disorders [24,31]. The multivariate analysis of complications by Heikkinen et al. [24] showed that secondary surgeries were characterized by up to a 9-fold increased risk of complications compared to all other operations performed on the thyroid gland (*p* < 0.001). Based on a multicenter study involving 16,448 patients, Dralle et al. showed that in the group of recurrent goiter patients, the risk of complications was almost 5-fold greater [31]. The selection of recurrent goiter surgery as a risk factor for complications should encourage surgeons and clinicians to carefully qualify patients for secondary thyroid surgery. In this case, assessing the risk of complications against the benefits that the patient can obtain after such surgery is a key element of the preoperative strategy.

One of the potential risk factors assessed was visual identification versus neuromonitoring. Identification of the RLN during thyroid surgery has been the accepted standard of care for years in order to avoid damage to the RLN [32,33] Therefore, the question arises whether neuromonitoring is a factor that protects the patient against complications, rather than only the visual identification used. We obtained results showing that the sole visualization of the nerve was associated with an almost two-fold higher risk of vocal cord paralysis (OR = 1.7; *p* = 0.02) compared to operations with neuromonitoring. This proves that laryngeal nerve monitoring should be considered a beneficial factor, minimizing the percentage of vocal cord paralysis. In 2009, Barczyński and Konturek showed in a randomized study that the use of neuromonitoring decreased the percentage of transient paralysis at the level of statistical significance. Another study involving 1000 RLNs exposed to the risk of injury showed that neuromonitoring did not significantly affect the risk of vocal cord paralysis in primary surgery, but it was significant in reducing the number of vocal cord paralysis cases in secondary surgery (19% vs. 7.8%) [34]. Dralle et al. [31] showed that in the coming years it will be difficult to prove the advantage of neuromonitoring over visual identification in the context of complications due to the lack of sufficient power of a statistical test, which should cover a minimum of 30,000 laryngeal nerves at risk of damage. The authors of many studies agree that this technique should be used in surgeries with an increased risk of complications [31,34,35,36]; this is also in line with the recommendations of the Polish Research Group for Neuromonitoring, which issued a statement in 2011 [37].

Summarizing the results of our analyses, it should be emphasized that identifying risk factors for complications of vocal cord paralysis is of particular importance for the practicing surgeon and for the center where these operations are performed. Knowledge of these factors may indicate thyroid surgery that requires monitoring of the laryngeal nerves to minimize the risk of damage to them.

The work presented above also has its limitations and weaknesses. Firstly, it is a retrospective study, which means that some data, especially regarding some risk factors, may have been imprecisely interpreted on the basis of available medical documentation (especially the assessment of retrosternal goiters, thyroid volume, or the presence and type of focal lesions in the thyroid gland). Secondly, thyroid surgeries were performed by different surgeons with different experience, in a selected period of time, which undoubtedly had an impact on the results of the research. Third, a complete assessment of phonation disorders after thyroid surgery should include assessment of both RLN damage and assessment of the EBSLN. Hung-Chun Chen et al., in their analysis of risk factors for phonation disorders, showed that one third of patients with unilateral vocal fold paralysis had accompanying damage to the external branch of the superior laryngeal nerve [14]. On the other hand, Jeannon et al. showed that 9 out of 20 patients operated on for medical conditions of the thyroid gland had postoperative damage to the EBSLN [1]. In the presented dissertation, due to the identification of the EBSLN at the level of only 20% in the center we represent, and the lack of an objective possibility to assess the damage to the EBSLN, the aspect of damage to this nerve was omitted. In these cases, where the EBSLN was identified we used several techniques to minimize the risk of injury to the EBSLN during superior thyroid vessel dissection and ligation. In the case of EBSLN neuromonitoring, cricothyroid muscle twitch was evaluated when the electromyographic waveform was higher than 100 μV. In the operation without neuromonitoring, we used several technics to prevent injury to the EBSLN. One of them is the ligation of the individual branches of superior thyroid vessels under direct vision on the thyroid capsule. Another is visual identification of the nerve before ligation of the superior thyroid pole vessels. In the future, a prospective cohort study should be planned, taking into account selected risk factors for damage to both laryngeal nerves, in order to fully assess phonation disorders.

## 5. Conclusions

In our study, we found that the presence of a retrosternal goiter, thyroid volume above 100 mL, male sex and secondary operations are the most important risk factors for phonation disorders after thyroidectomy. Thyroid cancer had a very large, statistically significant impact on the increase in permanent postoperative complications, while this indication had practically no impact on the transient paralysis. In patients over 65 years with obesity, the probability of transient paresis approximately doubled, with no effect on permanent vocal cord paralysis. The neuromonitoring of recurrent nerves should be considered a particularly advantageous factor, because operations with only visual identification of the laryngeal nerves were associated with a risk almost two times higher of both transient and permanent vocal cord paralysis.

## Figures and Tables

**Figure 1 biomedicines-10-02280-f001:**
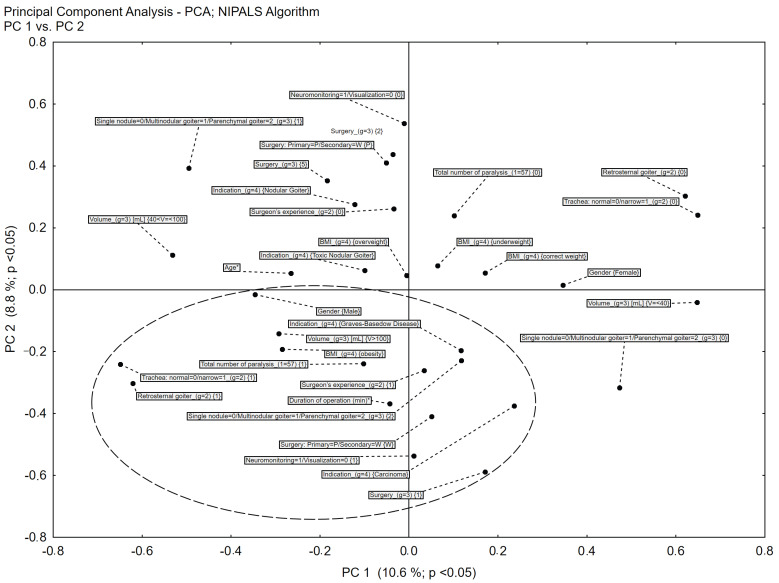
Principal component analysis of the relationship between risk factors and the total number of cases of vocal cord paralysis. Non-parametric tests for nominal variables based on Pearson’s chi2 statistics and the developed parametric logistic regression models confirmed the conclusions from the analysis of the results of the multidimensional data mining procedures.

**Figure 2 biomedicines-10-02280-f002:**
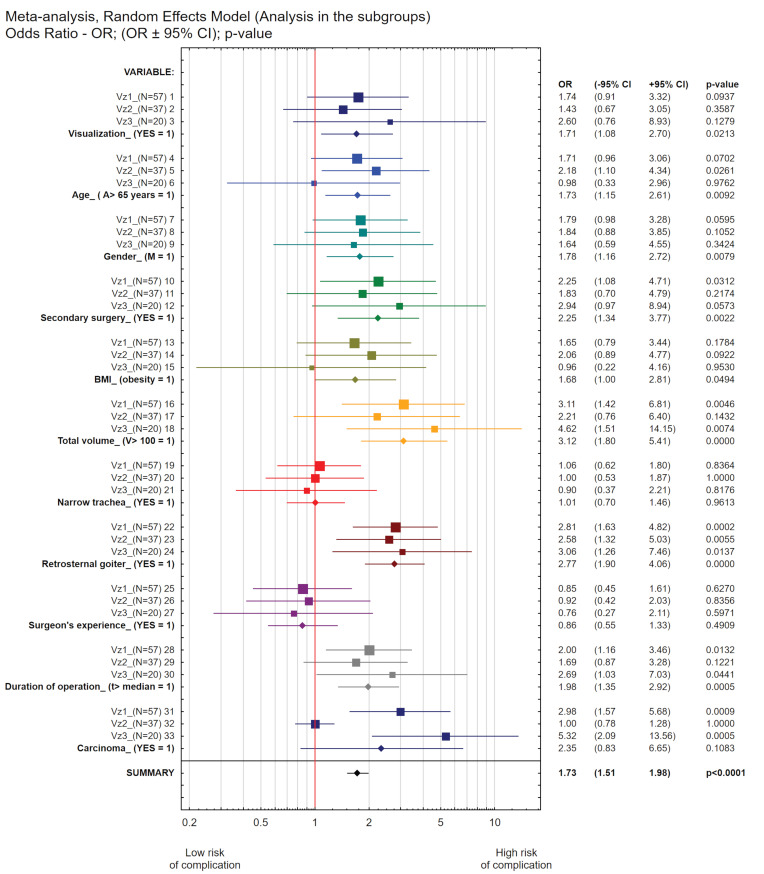
Meta-analysis of variable effects according to a scheme that takes into account all the possible interactions among the selected subgroups.

**Table 1 biomedicines-10-02280-t001:** Demographic data and clinical characteristics of patients treated surgically for thyroid diseases at the Department of General, Gastroenterological and Endocrine Surgery of the Medical University in Wrocław in 2011–2015 (*n* = 830 patients/1500 RLN/at risk).

Number of Patients, *n* (%)	830 (100%)
Number of RLN at risk of injury (RLN/at risk)	1500
Age, mean ± standard deviation Minimum/maximum age (years)	54.1 ± 14.2 17/86
Gender (F: M)	691:139
BMI, mean ± standard deviation (kg/m^2^) BMI minimum/maximum	26.2 ± 3.6 13/46
Tracheal displacement/constriction, *n* (%)	349 (42%)
Retrosternal goiter, *n* (%)	171 (21%)
Thyroid volume (V), mean ± standard deviation V minimum/maximum (ml)	42.2 ± 31.3 1/210
Primary surgery, *n* (%)	761 (92%)
Secondary surgery, *n* (%)	69 (8%)
Diagnosis, *n* (%)	
• Nodular goiter	591 (71.2%)
• Toxic nodular goiter	131 (15.8%)
Graves’ disease	32 (3.8%)
• Carcinoma	76 (9.2%)
− Papillary thyroid cancer − Follicular thyroid cancer − Medullary thyroid cancer − Anaplastic thyroid cancer	65 (7.8%) 8 (1.1%) 2 (0.2%) 1 (0.1%)
Thyroid surgery *n* (%)	
• Total thyroidectomy	495 (59.6%)
• Excision of a thyroid lobe with the isthmus	160 (19.3%)
• Almost total thyroidectomy	86 (10.4%)
• Dunhill procedure	45 (5.4%)
• Subtotal bilateral thyroidectomy	44 (5.3%)

RLN—recurrent laryngeal nerve, F—female, M—male, BMI—body mass index.

**Table 2 biomedicines-10-02280-t002:** Risk factors for complications of vocal cord paralysis and the number of RLNs at risk of injury.

Complication Risk Factor Symbol	Risk Factor for Complications of Vocal Cord Paralysis		Number of RLNs at Risk of Injury *n* = 1500 (100%)
*Vnz1*	RLN identification	Visual	1031 (69%)
		Neuromonitoring	469 (31%)
*Vnz2*	Age	≤65 y	1196 (80%)
		>65 y	304 (20%)
*Vnz3*	Gender	Female	1245 (83%)
		Male	255 (17%)
*Vnz4*	Thyroid surgery	Primary	1380 (92%)
		Secondary	120 (8%)
*Vnz5*	Trachea	Normal	863 (57%)
		Displaced/Constricted	638 (43%)
*Vnz6*	Retrosternal goiter	Absent	1179 (80%)
		Present	321 (20%)
*Vnz7*	Scope of thyroid surgery	Total thyroid lobectomy	1367 (91%)
		Partial thyroid lobectomy	133 (9%)
*Vzn8*	Surgeon/Experience	≤100 surgeries/year	304 (20%)
		>100 surgeries/year	80 (80%)
*Vnz9*	Clinical diagnosis	Nodular goiter	1045 (70%)
		Toxic nodular goiter	248 (16%)
		Graves’ disease	64 (4%)
		Carcinoma	143 (10%)
*Vzn10*	Focal lesion	Single	258 (17%)
		Multiple	1200 (80%)
		Parenchymal goiter	42 (3%)
*Vzn11*	Type of procedure	Total thyroidectomy	1195 (79.7%)
		Partial thyroid lobectomy	133 (8.8%)
		Almost total thyroidectomy	173 (11.5%)
*Vzn12*	BMI	Underweight	25 (2%)
		Normal	520 (35%)
		Overweight	799 (53%)
		Obesity	156 (10%)
*Vnz13*	Total volume	≤40 mL	964 (64%)
		>40 mL ≤100 mL	456 (31%)
		>100 mL	80 (5%)
*Vnz14*	Age–raw score	Age, mean ± standard deviation 54.1 ± 14.2 Minimum age–maximum 17–86	1500 (100%)
*Vnz15*	BMI—raw score	BMI, mean ± standard deviation (kg/m^2^) 26.2 ± 3.6, min-max 13/46	1500 (100%)
*Vnz16*	Volume—raw result	Thyroid volume * (V), mean ± standard deviation 42.2 ± 31.3 V minimum–maximum [mL] 1–210	1500 (100%)
*Vnz17*	Duration of operation—raw result	Time, mean ± standard deviation 58.3 ± 18.2 (min) Median 55 min	1500 (100%)

RLN—recurrent laryngeal nerve, BMI—body mass index. * The volume of the thyroid gland was calculated using simplified formula for the volume of the rotary ellipsoid: V = 0.5 × W × H × L, where V—the volume of the lobe, 0.5—simplified coefficient, W—width, H—thickness, L—length. The volume of the thyroid gland is the sum of the volumes of the right and left lobes.

**Table 3 biomedicines-10-02280-t003:** Meta-analysis of variable effects according to a scheme taking into account all the possible interactions among the selected subgroups (Vz1, Vz2, Vz3; Vz1 + Vz2 + Vz3).

**Complications Risk Factors**	**End Points**	OR	Standard Error	“−95% CI”	“+95% CI”	*p*	Meta-Analysis in Subgroups, % Share	OR	Standard Error	“–95% CI”	“+95% CI”	*p*	Share %
Visualization_ (YES = 1) Age_ (>65 years = 1) Gender_ (M = 1) Secondary surgery_ (YES = 1) BMI_ (obesity = 1) Total volume_ (V > 100 = 1) Narrow trachea_ (YES = 1) Retrosternal goiter_ (YES = 1) Surgeon’s experience_ (YES = 1) Duration of operation_ (t> median = 1) Indication: Carcinoma_ (YES = 1)	Vz1_ (*n* = 57)	1.74 1.71 1.79 2.25 1.65 3.11 1.06 2.81 0.85 2.00 2.98	0.57 0.51 0.55 0.85 0.62 1.24 0.29 0.77 0.28 0.56 0.98	0.91 0.96 0.98 1.08 0.79 1.42 0.62 1.63 0.45 1.16 1.57	3.32 3.06 3.28 4.71 3.44 6.81 1.80 4.82 1.61 3.46 5.68	0.0937 0.0702 0.0595 0.0312 0.1784 0.0046 0.8364 0.0002 0.6270 0.0132 0.0009	49.99% 50.10% 49.40% 49.28% 49.81% 49.16% 48.20% 49.37% 49.54% 49.87% 33.36%	1.83	0.23	1.44	2.33	0.0000	8.77% 9.94% 9.49% 7.38% 7.46% 6.78% 10.94% 10.78% 9.02% 10.63% 8.80%
Visualization_ (YES = 1) Age_ (> 65 years = 1) Gender_ (M = 1) Secondary surgery_ (YES = 1) BMI_ (obesity = 1) Total volume_ (V > 100 = 1) Narrow trachea_ (YES = 1) Retrosternal goiter_ (YES = 1) Surgeon’s experience_ (YES = 1) Duration of operation_ (t > median = 1) Indication: Carcinoma_ (YES = 1)	Vz2_ (*n* = 37)	1.43 2.18 1.84 1.83 2.06 2.21 1.00 2.58 0.92 1.69 1.00	0.55 0.76 0.69 0.90 0.88 1.20 0.32 0.88 0.37 0.57 0.13	0.67 1.10 0.88 0.70 0.89 0.76 0.53 1.32 0.42 0.87 0.78	3.05 4.34 3.85 4.79 4.77 6.40 1.87 5.03 2.03 3.28 1.28	0.3587 0.0261 0.1052 0.2174 0.0922 0.1432 1.0000 0.0055 0.8356 0.1221 1.0000	36.25% 35.95% 33.23% 29.02% 37.84% 26.74% 34.88% 32.37% 31.45% 33.94% 37.42%	1.49	0.19	1.16	1.92	0.0019	7.89% 9.05% 8.20% 5.53% 6.81% 4.70% 10.17% 9.38% 7.41% 9.46% 21.41%
Visualization_ (YES = 1) Age_ (>65 years = 1) Gender_ (M = 1) Secondary surgery_ (YES = 1) BMI_ (obesity = 1) Total volume_ (V > 100 = 1) Narrow trachea_ (YES = 1) Retrosternal goiter_ (YES = 1) Surgeon’s experience_ (YES = 1) Duration of operation_ (t> median = 1) Indication: Carcinoma_ (YES = 1)	Vz3_ (*n* = 20)	2.60 0.98 1.64 2.94 0.96 4.62 0.90 3.06 0.76 2.69 5.32	1.64 0.55 0.85 1.67 0.72 2.64 0.41 1.39 0.40 1.32 2.54	0.76 0.33 0.59 0.97 0.22 1.51 0.37 1.26 0.27 1.03 2.09	8.93 2.96 4.55 8.94 4.16 14.15 2.21 7.46 2.11 7.03 13.56	0.1279 0.9762 0.3424 0.0573 0.9530 0.0074 0.8176 0.0137 0.5971 0.0441 0.0005	13.76% 13.95% 17.37% 21.70% 12.35% 24.10% 16.92% 18.26% 19.01% 16.19% 29.22%	2.0	0.43	1.32	3.04	0.0011	7.50% 8.62% 9.43% 8.54% 5.87% 8.46% 10.78% 10.91% 9.44% 10.07% 10.37%
Summary—Effect of Interaction in Subgroups. Meta-Analysis in Subgroups													
Visualization_ (YES = 1) Age_ (>65 years = 1) Gender_ (M = 1) Secondary surgery_ (YES = 1) BMI_ (obesity = 1) Total volume_ (V > 100 = 1) Narrow trachea_ (YES = 1) Retrosternal goiter_ (YES = 1) Surgeon’s experience_ (YES = 1) Duration of operation_ (t > median = 1) Indication: Carcinoma_ (YES = 1)	Vz1 + Vz2 + Vz3 *n* = 57 + 37 + 20	1.71 1.73 1.78 2.25 1.68 3.12 1.01 2.77 0.86 1.98 2.35	0.40 0.36 0.39 0.59 0.44 0.88 0.19 0.54 0.19 0.39 1.25	1.08 1.15 1,16 1.34 1.00 1.80 0.70 1.90 0.55 1.35 0.83	2.70 2.61 2.72 3.77 2.81 5.41 1.46 4.06 1.33 2.92 6.65	0.0213 0.0092 0.0079 0.0022 0.0494 0.0000 0.9613 0.0000 0.4909 0.0005 0.1083	100.00% 100.00% 100.00% 100.00% 100.00% 100.00% 100.00% 100.00% 100.00% 100.00% 100.00%						
Summary		1.73	0.12	1.51	1.98	0.0000	-						

M—male, BMI—body mass index, Vz1—total number of paralysis cases: transient and permanent, Vz2—transient paralysis, Vz3—permanent paralysis, *p*—statistical significance *p* <0.05.

## Data Availability

The datasets used and/or analyzed during the current study are available from the corresponding author upon reasonable request.

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
