# Peer review of "Analysis of Risk Factors for Phonation Disorders after Thyroid Surgery"

_biomedicines, 2022, doi:10.3390/biomedicines10092280_

Round 1
Reviewer 1 Report
In this retrospective study the authors conducted a multivariate analyses on 830 patients who underwent surgery on 1500 recurrent laryngeal nerves. They identified several risk factors that were associated with either temporary or permanent paralysis. There are several comments:
Major comments:
1. Data Collection. Vz1-3 is defined as immediate postoperative complication. However, in the immediate postoperative period it yet unknown if the paralysis will be transient or permanent.
2. How were the vocal cords assessed? Were all patients subject to fiberoptic examination of the larynx, or a validated questionnaire was used? Was a video-stroboscope used in each examination? When was the laryngeal examination scheduled? Was it performed in the immediate postop. period, or in the clinic?
3. What formula was used to calculate the thyroid volume? Was it calculated out of the CT scans or from the pathology report?
4. Please elaborate on the technique used to control the superior pole vessels, a it is important for the preservation of the EBSLN.
5. Were all retrosternal goiters removed transcervical?
6. Table 2 can be presented in a sentence or two.
7. According to Table 1 19.3% of cases were a hemithyroidectomy and the rest were some sort of surgery that put at risk nerves at both sides of the neck. However, in Table 3, Vnz7 describes partial thyroidectomy 9% and total thyroidectomy 91%. Can you please explain.
8. Did all patients have a fiberoptic laryngeal examination before surgery? How many patients had a vocal cord paralysis preoperatively?
9. Section 3.1 Page 7. Please include 95%CI when presenting OR data.
10. P8L72. Please change “on the verge of” to “had no”, and remove OR, and Chi2 as they did not reach statistical significance.
11. When presenting OR in the text please include 95%CI as well.
12. Discussion. P15L433. What drugs make the nerve more prone to damage?
13. Discussion. P15L460. If gender caused delayed diagnosis and as a consequence a retrosternal goiter, than it would be apparent in the multivariate analyses.
14. Conclusions – there is no need to number each finding. Try to be more concise.
Minor comments:
1. Abstract. P1L18 an ‘and’ is missing before ‘thyroid’.
2. Abstract P1L18 “We found that” before “The retrosternal goiter”.
3. Abstract: P1. All results should be reported in past tense.
4. Introduction. P1L34. “and” before “singers”.
5. Introduction. P2L48, a reference number is missing in the end of the line.
6. Introduction. P2L52 “is” before “responsible”.
7. Data Collection. P3L90 “is presented”.
8. Table 2: You misplaced TP (transient) and PP (permanent).
9. When using abbreviation in tables and figures – please add their explanation in the description below.
10. Table 4. Instead of 3 columns for OR and 95%CI it is usually presented in one column as: OR 95CI (upper-lower).
11. P13L332 “may cause”.
12. P13L336 replace “material” with “cohort”.
13. Discussion. P15L423. A period is missing after 1.7.
14. Discussion. P16L512 “our” not “my”.
15. P17L530. “we” not “I”.
Author Response
Dear Reviewer,
Thank you for you insightful, valuable corrections and time for reviewing the manuscript. I made the changes as best I could.
Yours sincerely,
Authors
Review 1
In this retrospective study the authors conducted a multivariate analyses on 830 patients who underwent surgery on 1500 recurrent laryngeal nerves. They identified several risk factors that were associated with either temporary or permanent paralysis. There are several comments:
Major comments:
- Data Collection. Vz1-3 is defined as immediate postoperative complication. However, in the immediate postoperative period it yet unknown if the paralysis will be transient or permanent.
Of course, I agree with the reviewer’s opinion that in the immediate postoperative period it is yet unknown if the paralysis will be transient or permanent. The aim of the study was to evaluate three points: 1) the total number of vocal fold paralysis, which is the sum of transient and permanent paresis, 2) transient paresis and 3) permanent paralysis. The sentence used was imprecise and could be misleading. The sentence was deleted and the 3 endpoints were clearly identified.
Actual version is:
All of them were early postoperative complications observed in the immediate postoperative period: Vz1 included both temporary and permanent paralysis, Vz2 was transient paralysis and Vz3 was permanent paralysis. The total number of vocal cord paralysis (Vz1) after thyroid surgery was 57 (3.8%) calculated on the number of RLN at risk of injury (1500 RLN at risk): transient paralysis (TP/Vz2) - 37 (2.5%) and permanent paralysis (PP/Vz2)- 20 (1.3%). Unilateral paresis occurred in 49 (5.9%) patients and bilateral paresis in 4 (0.5%) patients.
- How were the vocal cords assessed? Were all patients subject to fiberoptic examination of the larynx, or a validated questionnaire was used? Was a video-stroboscope used in each examination? When was the laryngeal examination scheduled? Was it performed in the immediate postop. period, or in the clinic?
Before each operation the patients underwent ENT examination of the vocal cords (indirect examination or videolaryngoscopy). The same examination was also performed on the first or second postoperative day. In cases of abnormal vocal cord mobility, videostroboscopy was performed to confirm paresis or paralysis of the vocal cords and to assess the glottis width. The examination of vocal cords was repeated after 1, 3 and 6 months in cases of proven paralysis. Transient paralysis was defined as paralysis that resolved within 6 months after surgery; if the paralysis persisted for 6 months it was classified as permanent - this information has been added to the section Material and Methods.
- What formula was used to calculate the thyroid volume? Was it calculated out of the CT scans or from the pathology report?
Thyroid volume was calculated from data collected on the basis of thyroid ultrasound. Only a few patients had a CT scan in addition to the thyroid ultrasound. The volume of the thyroid gland was calculated using simplified formula for the volume of the rotary ellipsoid: V=0.5 x W x H x L, where V - the volume of the lobe, 0.5 - simplified coefficient, W - width, H - thickness, L - length The volume of the thyroid gland is the sum of the volumes of the right and left lobe. Below the Table 3 the following formula was added : (*The volume of the thyroid gland was calculated using simplified formula for the volume of the rotary ellipsoid: V=0.5 x W x H x L, where V- the volume of the lobe, 0.5 - simplified coefficient, W - width, H - thickness, L - length. The volume of the thyroid gland is the sum of the volumes of the right and left lobe).
- Please elaborate on the technique used to control the superior pole vessels, a it is important for the preservation of the EBSLN.
In the case of EBSLN neuromonitoring, CTM (cricothyroid muscle) twitch was evaluated. For both RLN and EBSLN, positive signal/stimulation was recognizable when the electromyographic waveform was higher than 100 μV. In the operation without neuromonitoring we use several technics to prevent EBSLN. One of them is the ligation of the individual branches of superior thyroid vessels under direct vision on the thyroid capsule. Another is visual identification of the nerve before ligation of the superior thyroid pole vessels. In the end of the discussion this explanation was found.
- Were all retrosternal goiters removed transcervical?
Yes, all patients with retrosternal goiters underwent cervical access surgery.
- Table 2 can be presented in a sentence or two.
Table 2 has been canceled as suggested and replaced with the following text:The total number of vocal cord paralysis (Vz1) after thyroid surgery was 57 (3.8%) calculated on the number of RLN at risk of injury (1500 RLN at risk): transient paralysis (TP/Vz2) - 37 (2.5%) and permanent paralysis (PP/Vz2)- 20 (1.3%). Unilateral paresis occurred in 49 (5.9%) patients and bilateral paresis in 4 (0.5%) patients.
- According to Table 1 19.3% of cases were a hemithyroidectomy and the rest were some sort of surgery that put at risk nerves at both sides of the neck. However, in Table 3, Vnz7 describes partial thyroidectomy 9% and total thyroidectomy 91%. Can you please explain.
We assessed the paralysis on the RLN at risk. The potential risk factor was the scope of the operation- that is, we could remove the lobe of the thyroid completely or partially. We have:
- Total thyroidectomy (495 patients/ 990 RLNs at risk= 990 lobes of the thyroid removed completely)
- Hemithyroidectomy (160 patients/160 RLNs at risk= 160 lobes of the thyroid removed completely)
- Almost total thyroidectomy - 86 patients/172 RLNs at risk= 172 lobes of the thyroid removed completely (It was such a kind of the operation where we left less than 10 mm on both sides during operation - this kind of procedures in our Department is similar to total thyroidectomy)
- Dunhill procedure 45 patients/ 90 RLNs at risk= 45 lobes removed completely and 45 lobes removed subtotal/partial
- Subtotal bilateral thyroidectomy 44 patients/ 88 RLNs at risk= 88 lobes removed subtotal/partial
To sum up we have 45+88=133 (9%) lobes removed subtotal
- Did all patients have a fiberoptic laryngeal examination before surgery? How many patients had a vocal cord paralysis preoperatively?
Yes, before each operation the patients underwent ENT examination of the vocal cords (indirect examination or videolaryngoscopy). Of course, sometimes we have patients with the vocal cords paralysis f.e. after primary thyroid operation, but these patients were not included in the study.
- Section 3.1 Page 7. Please include 95%CI when presenting OR data.
(chi2 = 15.1; p = 0.0001, OR 2.8 , 95%CI 1.63-4.82).
(chi2 = 4.9; p = 0.027, OR 2.25, 95%Cl 1.08-4.71)
- P8L72. Please change “on the verge of” to “had no”, and remove OR, and Chi2 as they did not reach statistical significance.
I have changed “on the verge of” to “had no” and cancelled OR and Chi2.
- When presenting OR in the text please include 95%CI as well.
- Discussion. P15L433. What drugs make the nerve more prone to damage?
I added as following: ….drugs such as: antithrombotic and antiplatelet …
- Discussion. P15L460. If gender caused delayed diagnosis and as a consequence a retrosternal goiter, than it would be apparent in the multivariate analyses.
I added …. but subjective….
Our clinical, but subjective experience indicates that thyroid surgery performed on men can be technically more difficult than on women.
- Conclusions – there is no need to number each finding. Try to be more concise.
My proposal is to change this part of manuscript, actual version is below:
Retrosternal goiter, thyroid’s volume above 100 ml, male sex, secondary operations are the most important risk factors for phonation disorders after thyroidectomy. Thyroid cancer has a very large, statistically significant impact on the increase of permanent postoperative complications, while this indication has practically no impact on the transient paralysis. In patients over 65 years with obesity the probability of transient paresis approximately double, with no effect on permanent vocal cord paralysis. The neuromonitoring of recurrent’s nerves should be considered a particularly advantageous factor, because operations with only visual identification of the laryngeal nerves were associated with an al-most twice higher risk of both transient and permanent vocal cord paralysis.
Minor comments:
- Abstract. P1L18 an ‘and’ is missing before ‚thyroid’.
Ok, the word „and" has been added before „thyroid” in P1L18.
- Abstract P1L18 “We found that” before “The retrosternal goiter”.
Ok,“We found that” has been added before “The retrosternal goiter” in P1L18
- Abstract: P1. All results should be reported in past tense.
In the abstract, the present tense I have changed in past tense: („are” in „were” P1L19; „has” in „had” P1L20, „has” in „had” P1L21; „are” in „were” P1L23.
- Introduction. P1L34. “and” before “singers”. I added “and” before “singers” .in P1L34
- Introduction. P2L48, a reference number is missing in the end of the line. I put a reference number [1] in P2L48.
- Introduction. P2L52 “is” before “responsible”. I put „is” before „responsible” in P2L52.
- Data Collection. P3L90 “is presented”. It was changed in P3L90.
- Table 2: You misplaced TP (transient) and PP (permanent). Of course, I have corrected Table 2; now is Transient paralysis (TP) and Permanent paralysis (PP).
- When using abbreviation in tables and figures – please add their explanation in the description below.
Below the Table 1 I have added their explanations (RLN - recurrent laryngeal nerve, F - female, M - male, BMI - Body Mass Index.
Below the Table 2 I have added their explanations (RLN - recurrent laryngeal nerve,, BMI - Body Mass Index.
In table 4 : Age_ (W> 65 years = 1) was changed into Age_ (> 65 years = 1)
Below the Table 4 I have added: M - male, BMI - Body Mass Index, Vz1 - Total number of paralysis: transient and permanent, Vz2 - transient paralysis Vz3 - permanent paralysis, p - statistic significance <0.05
- Table 4. Instead of 3 columns for OR and 95%CI it is usually presented in one column as: OR 95CI (upper-lower).
- P13L332 “may cause”. The word „cause’ I have changed into „may cause’. P13L332.
- P13L336 replace “material” with “cohort”. The word „material” was replaced with „cohort” in P13L336.§
- Discussion. P15L423. A period is missing after 1.7. Ok, the period has been added to the end of the sentence - P15L423.
- Discussion. P16L512 “our” not “my”. The word „my” was changed into „our” in P16L512.
- P17L530. “we” not “I”. The word „I” was changed into „we” in P17L530.
Please see the attachment.

Reviewer 2 Report
The authors presented an interesting report n the incidence and risk factors for phonation disorders after thyroid surgery. The authors performed a hard work in presenting their data, but some points should be improved: how was the thyroid volume calculated? Did you use the ellipsoid formula or any other? Did you perform a laryngological examination in all patients? This could an important bias of the study, since some of the phonation disorders could be already present at the time of surgery.
In the results section reference to statistical analysis should be reduced: the explanation of every single OR value is not always necessary and it should be sufficient to indicate the value.
Author Response
Dear Reviewer,
Thank you for insight valuable corrections and time for reviewing the manuscript I made the changes as best I could.
Yours sicerely,
Beata Wojtczak
Review 2
The authors presented an interesting report n the incidence and risk factors for phonation disorders after thyroid surgery. The authors performed a hard work in presenting their data, but some points should be improved: how was the thyroid volume calculated? Did you use the ellipsoid formula or any other?
Did you perform a laryngological examination in all patients?
This could an important bias of the study, since some of the phonation disorders could be already present at the time of surgery.
In the results section reference to statistical analysis should be reduced: the explanation of every single OR value is not always necessary and it should be sufficient to indicate the value.
Thyroid volume was calculated from data collected on the basis of thyroid ultrasound. Only a few patients had a CT scan in addition to the thyroid ultrasound. The volume of the thyroid gland was calculated using simplified formula for the volume of the rotary ellipsoid: V=0.5 x W x H x L, where V - the volume of the lobe, 0.5-simplified coefficient, W - width, H - thickness, L – length. The volume of the thyroid gland is the sum of the volumes of the right and left lobe. Below the Table 3 the following formula was added : (*The volume of the thyroid gland was calculated using simplified formula for the volume of the rotary ellipsoid: V=0.5 x W x H x L, where V - the volume of the lobe, 0.5 - simplified coefficient, W - width, H - thickness, L – length. The volume of the thyroid gland is the sum of the volumes of the right and left lobe).
Before each operation the patients underwent ENT examination of the vocal cords (indirect examination or videolaryngoscopy). The same examination was also performed on the first or second postoperative day. In cases of abnormal vocal cord mobility, videostroboscopy was performed to confirm paresis or paralysis of the vocal cords and to assess the glottis width. The examination of vocal cords was repeated after 1, 3 and 6 months in cases of proven paralysis. Transient paralysis was defined as paralysis that resolved within 6 months after surgery; if the paralysis persisted for 6 months it was classified as permanent - this information has been added to the section Material and Methods.

Round 2
Reviewer 1 Report
Major comment #11 was not addressed:
- When presenting OR in the text please include 95%CI as well.
Author Response
Dear Reviewer,
Sorry to miss the point, all the missing %95 Cl ( OR) are filled . I marked the changes in blue in the text.
Kind regards
Beata Wojtczak

Reviewer 2 Report
The authors addressed all my comments
Author Response
Dear Reviewer,
Thank you for your time and commitment.
Kind regards
Beata Wojtczak

Round 3
Reviewer 1 Report
The authors have addressed all comments.